# Epitaxial Graphene on 4H-SiC (0001) as a Versatile Platform for Materials Growth: Mini-Review

**Ivan Shtepliuk** [1,*], **Filippo Giannazzo** [2] **and Rositsa Yakimova** [1]

1   Department of Physics Chemistry and Biology, Linköping University, SE-58183 Linköping, Sweden; rositsa.yakimova@liu.se
2   CNR-IMM, Strada VIII, 5, 95121 Catania, Italy; Filippo.Giannazzo@imm.cnr.it
*   Correspondence: ivan.shtepliuk@liu.se; Tel.: +46-766-524-089

**Abstract:** Material growth on a dangling-bond-free interface such as graphene is a challenging technological task, which usually requires additional surface pre-treatment steps (functionalization, seed layer formation) to provide enough reactive sites. Being one of the most promising and adaptable graphene-family materials, epitaxial graphene on SiC, due to its internal features (substrate-induced *n*-doping, compressive strain, terrace-stepped morphology, bilayer graphene nano-inclusions), may provide pre-conditions for the enhanced binding affinity of environmental species, precursor molecules, and metal atoms on the topmost graphene layer. It makes it possible to use untreated pristine epitaxial graphene as a versatile platform for the deposition of metals and insulators. This mini-review encompasses relevant aspects of magnetron sputtering and electrodeposition of selected metals (Au, Ag, Pb, Hg, Cu, Li) and atomic layer deposition of insulating $Al_2O_3$ layers on epitaxial graphene on 4H-SiC, focusing on understanding growth mechanisms. Special deliberation has been given to the effect of the deposited materials on the epitaxial graphene quality. The generalization of the experimental and theoretical results presented here is hopefully an important step towards new electronic devices (chemiresistors, Schottky diodes, field-effect transistors) for environmental sensing, nano-plasmonics, and biomedical applications.

**Keywords:** epitaxial graphene; SiC; deposition; metals; insulators; growth mechanism



## 1. Introduction

The achievement of large area homogeneous monolayer graphene epitaxially grown on the Si face of SiC (0001) [1] has allowed the rapid development of the new class of 2D materials in line with exfoliated graphene and boosted some exciting applications in quantum metrology [2–5], and environmental sensing [6,7], which might not be possible using free-standing graphene. It has been demonstrated that epitaxial graphene on SiC, because of its unique structure, distinct from that of other graphene family representatives [8–10], is a promising material to design the next-generation quantum resistance standard of high precision, which make it possible to calibrate mass. As a breakthrough outcome, recently, the electronic kilogram has been introduced. During the last decade, large efforts have been made to develop epitaxial graphene-based sensors for the detection of $NO_2$ [11–13], polar protic and polar aprotic vapors [14], solvents [15,16], Hall effect [17,18], illicit drugs (amphetamine or cocaine) [19], toxic heavy metals (cadmium, lead, and mercury) [20–25], ultraviolet (UV) light [26], and terahertz waves [27], etc. Notably, the above-listed examples of sensing applications concern only unmodified pristine epitaxial graphene. At the same time, there is a great deal of interest in epitaxial graphene sensitivity improvement by chemical functionalization [28,29], defect engineering [30,31], and metal oxide nanoparticle decoration [32–34].

Applicability and functionality of epitaxial graphene are attributable, in the first instance, to its transfer-free growth on a native substrate through the Si sublimation process during high-temperature annealing of the SiC substrate in an argon atmosphere. The

presence of contact between the substrate and graphene is mainly responsible for the observed *n*-doping and compression of the topmost graphene layer [35–37], which is, however, beneficial to promotion of higher reactivity of epitaxial graphene compared to exfoliated counterpart. Indeed, the compressive strain can significantly influence the adsorption capability of graphene with respect to environmental species and individual atoms/molecules [38,39], while electron doping is regarded as a viable means to increase the reactivity toward oxygen [40]. This creates good prerequisites not only for boosting innovative sensing and catalytic applications but also for the exploitation of epitaxial graphene as a substrate for the growth of other functional materials, such as extremely thin metal layers and nanoparticles, semiconducting films, and insulators with good adhesion. The latter is of relevance in the light of the need to expand the functionality of the epitaxial graphene towards nano-plasmonic sensors for environmental detection/monitoring and field-effect transistor (FET)-based biosensors.

Although the fundamental properties of epitaxial graphene and physics behind epitaxial-graphene-based devices are well documented in the scientific literature, further progress in implementing epitaxial graphene into realistic applications requires both rethinking of the existing knowledge in this field and systematization of new experimental and theoretical results. The current paper is intended to provide a focused overview of the recent progress in the investigation of the deposition of selected metals and insulators on epitaxial graphene supported by 4H-SiC. Direct current (dc) magnetron sputtering, electro-chemical deposition and atomic layer deposition (ALD) methods for thin layer synthesis are presented and discussed. We also highlight the nature of the interrelationships between epitaxial graphene quality and the morphology of deposits. In this mini-review paper, we mainly focus on the generalization of recent studies performed at Linköping University (Linköping, Sweden) and CNR-IMM (Catania, Italy).

## 2. Key Properties of Epitaxial Graphene as a Substrate for Materials Deposition

Before delving into the discussion of issues related to deposition of metals and insulators on epitaxial graphene on 4H-SiC, we would like first to recall the peculiar properties of epitaxial graphene that make it appealing to the growth of diverse functional materials. For all cases of the materials growth, which will be considered at a later stage, the common feature is that we used the same type of substrate: monolayer epitaxial graphene (MLG) prepared by high-temperature (~1900 °C) thermal decomposition of the Si-face of nominally on-axis 4H-SiC (0001) substrates in Ar atmosphere [41]. The graphene formation process encompasses three successive stages: (i) initial Si sublimation that takes place in inductively heated graphite enclosure in highly controlled conditions (gas, pressure, time), (ii) C-rich $6\sqrt{3} \times 6\sqrt{3}$ R30° surface reconstruction of (0001) face of 4H-SiC (buffer layer formation), and (iii) final formation of topmost graphene layer above the buffer layer. The key stages behind MLG synthesis are illustrated in Figure 1.

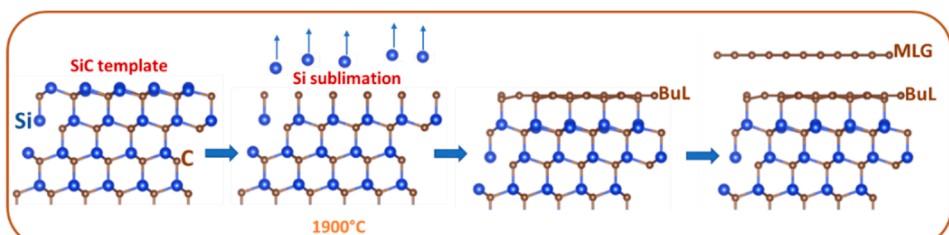

**Figure 1.** The experimental steps leading to the formation of monolayer epitaxial graphene on SiC. This scheme has been adapted with permission from [42]. Copyright 2021 The Authors. Published by American Chemical Society (ACS), Washington, United States. Further permissions related to the material excerpted should be directed to the ACS, Washington, United States.

MLG quality control is carried by two independent methods: optical reflectance mapping [43] and micro-Raman spectroscopy. Typically, the grown samples are composed

of 85% monolayer graphene (and even larger % for recently synthesized samples) and 15% bilayer graphene that is represented by small bilayer patches above the first graphene layer (Figure 2a). The presence of randomly distributed bi-layer graphene islands within MLG sample can be beneficial to provide extra nucleation sites during early growth stages. More information about the intrinsic properties of the epitaxial graphene on SiC can be extracted from the analysis of the Raman mapping data [42].

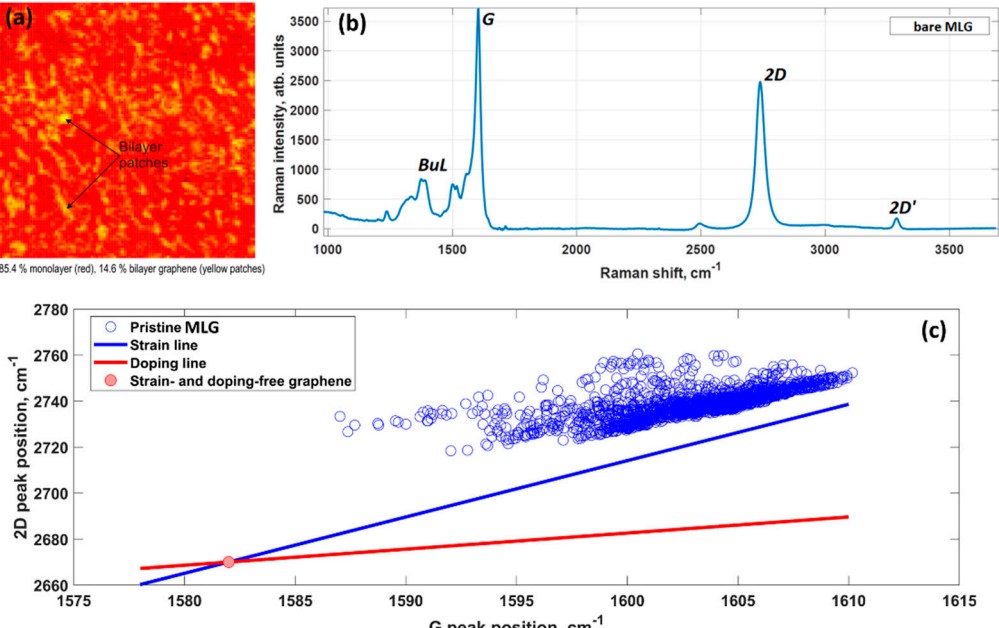

**Figure 2.** (**a**) The optical reflectance map confirming the prevalence of monolayer graphene on a $30 \times 30 \ \mu m^2$ area. Bilayer patches are seen in yellow. (**b**) Raman spectrum of typical monolayer epitaxial graphene sample. *BuL* represents the spectral features related to the buffer layer. (**c**) Dependence of 2*D* peak position on *G* peak position. The above figures have been adapted with permission from [42]. Copyright 2021 The Authors. Published by American Chemical Society, Washington, United States. Further permissions related to the material excerpted should be directed to the ACS, Washington, United States.

First, it was determined that the grown samples were defect-free, which was confirmed by the absence of defect-related Raman peaks (mainly *D* Raman modes, Figure 2b). Therefore, one may exclude defects (carbon vacancies or $sp^3$ defects) as preferential nuclei. Second, the experimental points in 2*D*–*G* space scattered primarily along the strain line with a slope of 2.4 (Figure 2c), which is indicative of *n*-doping and biaxial compression of the topmost graphene layer [43,44]. Density functional theory (DFT) calculations showed that the charge transfer between SiC and graphene caused the shift of the Dirac cone (0.49 eV) below the Fermi level $E_F$ (Figure 3), thereby corroborating experimental findings on SiC-substrate-induced *n*-type doping of graphene. In this regard, a comparison of the charge redistribution map of free-standing graphene with that of epitaxial graphene revealed a pronounced accumulation of negative charge on SiC-supported graphene sheet (Figure 4). As was shown by Schilirò et al. [44], there is a direct correlation between the electron density of graphene and the adsorption energy of water molecule. More particularly, the downshift of the Dirac point below the Fermi level is responsible for the adsorption energy increasing from 127 to 210 meV (Figure 5). It was argued that this phenomenon can be beneficial for the ALD process underlying the growth of $Al_2O_3$ layers using trimethyl aluminium (TMA) as the Al precursor and $H_2O$ as the co-reactant. Indeed, the better wettability of graphene by water, the larger number of nuclei for $Al_2O_3$ formation.

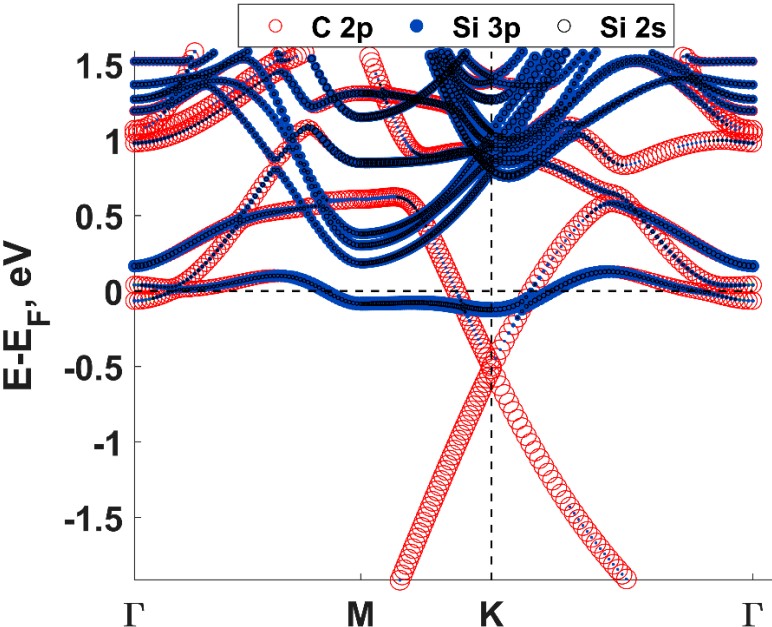

**Figure 3.** Fat band structure (calculated in the frames of the current work) for monolayer epitaxial graphene on 4H-SiC, including contributions of key orbitals to the bands. The size of each circle is proportional to the corresponding orbital contribution to each band. The Fermi level was set to 0 eV. The band-structure calculations were performed using $32 \times 32 \times 1$ k-point Monkhorst-Pack mesh at vdW/BH/DZP level of DFT by SIESTA code.

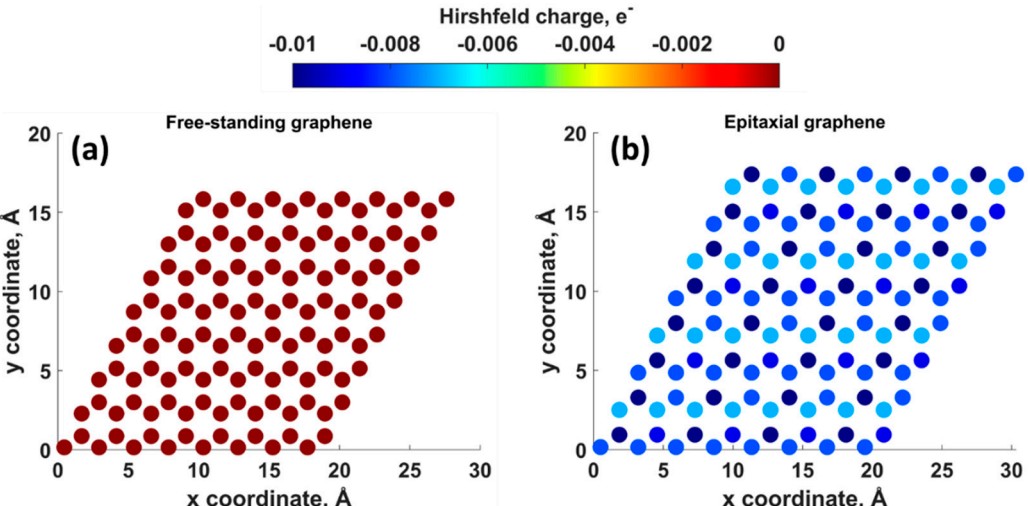

**Figure 4.** Charge redistribution maps (constructed based on Hirshfeld charges) of (**a**) free-standing graphene and (**b**) monolayer epitaxial graphene on 4H-SiC structure. The charge population analysis was performed within this work using $8 \times 8$ slabs (free-standing graphene and epitaxial graphene on SiC, respectively) at vdW/BH/DZP level of DFT by SIESTA code.

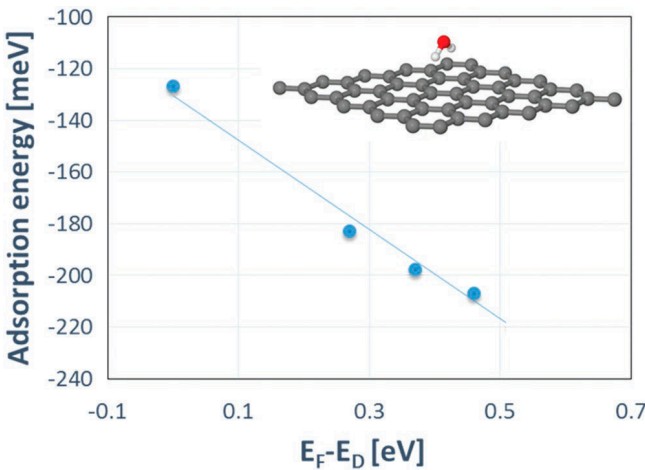

**Figure 5.** DFT-predicted adsorption energy of a $H_2O$ molecule on monolayer graphene versus $(E_F - E_D)$ values [44]. Note: The case when $(E_F - E_D)$ corresponds to neutral graphene, for which Fermi level coincides with the Dirac point. While $(E_F - E_D)$ value of 0.45 eV is related to *n*-type doping of $\approx 1.5 \times 10^{13}$ cm$^{-2}$. Copyright 2019 WILEY-VCH Verlag GmbH & Co. KGaA, Weinheim, Germany.

The analysis of Figure 2c also indicates that as-grown epitaxial graphene samples were under compressive strain. This is contrary to the results of theoretical calculations, according to which graphene on SiC would undergo a lattice expansion rather than compression. This is because the lattice constant of graphene is smaller than that of SiC. In line with this, our estimations showed that the topmost graphene layer experiences 9% tensile strain. This discrepancy can be explained by the fact that graphene on SiC is forced to compensate for the mismatch in their coefficients of thermal expansions through building up compressive stress after cooling down to room temperature [45].

While the adsorption capability of expanded graphene originates from the formation of localized dangling-bond states that can promote strong binding with environmental adsorbates [46], the mechanism behind the interaction between compressively strained graphene and environmental species is not so obvious. On the one hand, a shrinkage of graphene (C-C bond length shortening) may induce delocalization of electron wavefunction over the planar graphene structure and decrease the density of dangling bond states, which are available for surface reactions [46]. On the other hand, the compressive strain can cause the formation of nano-ripples, which are able to adsorb atomic species at ridge sites [38].

Further, we highlight four most probable highly reactive regions on epitaxial graphene/SiC, which may serve as native seeds for the growth of external materials:

(a) Step edges. Graphene on SiC has terrace-stepped morphology. Adsorption of atomic species on the steps edges is expected to be much stronger than on terraces.

(b) Charge-unbalanced electron-hole puddles. The supporting SiC substrate is responsible not only for the doping of graphene but also for the appearance of charge density fluctuations. Randomly distributed negatively and positively charged regions may favor increased adsorption and nucleation rate.

(c) Strained regions. Compressive or tensile strain leading to alteration of orbital hybridization and substrate-induced strain fluctuations are considered as another important factor determining an increased reactivity of epitaxial graphene compared to exfoliated graphene.

(d) Edges of the bilayer graphene inclusions. Since as-grown monolayer epitaxial graphene usually includes nanoscale overgrown areas related to bilayer graphene, it is reasonable to assume that unsaturated edges of these inclusions may create extra nucleation sites.

## 3. Deposition of Noble Metals on Epitaxial Graphene/4H-SiC by dc Magnetron Sputtering

The behavior of noble metals on epitaxial graphene is a matter of considerable interest, as its deep understanding may boost the development of new sensor designs based on nano-plasmonics. Our recent findings shed light on the interplay between two selected magnetron-sputtered noble metals (silver and gold as representatives of metals with best plasmonic activity) and epitaxial graphene on 4H-SiC [43,47–49]. From the theoretical point of view, both considered metals interact with epitaxial graphene through weak van der Waals forces, which is confirmed by a low adsorption energy that is below the lower limit of chemisorption and small charge transfer from metals to graphene [49]. Concomitantly, silver atoms interact stronger with graphene surfaces compared to gold species. A direct comparison of 2 nm-Ag and 2nm-Au MLG samples showed that the difference in the interaction strengths translates into a difference in morphologies (Figure 6) and Raman responses of graphene. More specifically, gold on epitaxial graphene forms fractal-like nanostructures (coverage area of 36.6%) with a fractal dimension of 1.84 [42], which was due to limited atomic diffusion at island edges. At the same time, 2nm-silver film consists of small, isolated nano-islands (average island diameter of ~15 nm; coverage area of 32.2%). Such a growth distinction is mainly driven by the difference in the interrelationships between the adsorption energy and the cohesive energy of gold and silver supported by epitaxial graphene. According to our DFT calculations (Figure 7), both metals tend to occupy the hollow site positions of the topmost graphene layer at different heights (2.07 Å and 2.23 Å for Ag and Au, respectively), demonstrating a clear difference in the adsorption energies (0.355 eV for Ag vs. 0.255 eV for Au).

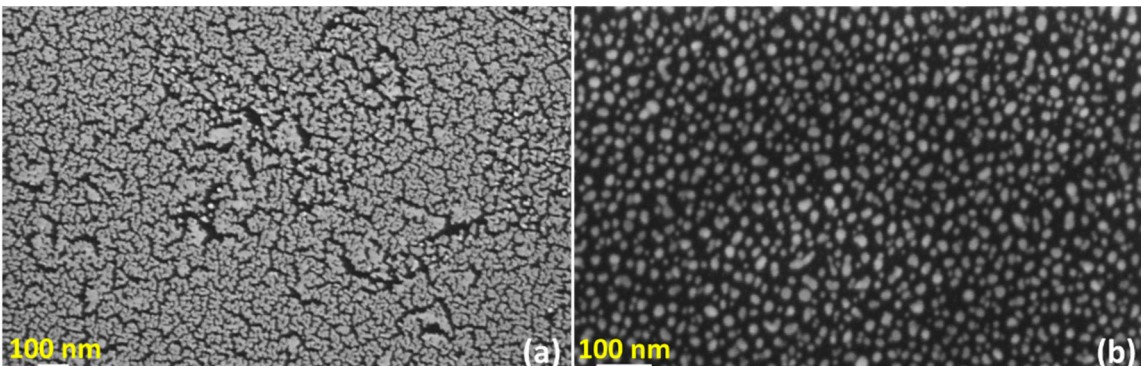

**Figure 6.** SEM images of (**a**) 2nm-Au and (**b**) 2 nm-Ag films deposited on epitaxial graphene. Copyright 2021 WILEY-VCH Verlag GmbH & Co. KGaA, Weinheim, Germany.

The fact that the adsorption energy of gold is 100 meV smaller than that of silver presupposes a much smaller probability for Au adatoms to be trapped by an epitaxial graphene surface to form early-stage nuclei in comparison to Ag adatoms. Furthermore, the predicted cohesive energy of bulk gold has a more negative value (−3.62 eV) as compared to the cohesive energy of bulk silver (−2.95 eV), suggesting that the Au-Au interaction is more stable and stronger than the Ag-Ag interaction. This difference provides an additional driving force for the MLG-assisted self-assembly of just adsorbed gold into interconnected nanostructures. Back to the silver case, the higher adsorption energy and the less negative cohesive energy seem to be sufficient factors to ensure a high density of small, separated Ag islands. It is interesting to note that the difference in the interaction strength characteristically manifests itself in Raman spectra of epitaxial graphene (Figure 8) [49]. Particularly, the Raman spectrum of pristine epitaxial graphene remains typically intact after gold deposition (except for small displacements of the main characteristic *G* and 2*D* peaks), while silver deposition causes substantial changes in Raman activity reflected in an appearance of defect-related Raman peaks (*D*, *D'*, *D* + *D'*, *D* + *G*, and *D* **) and huge red-shift of the 2*D* peak. The latter can be attributed to the *n*-doping of graphene [48].

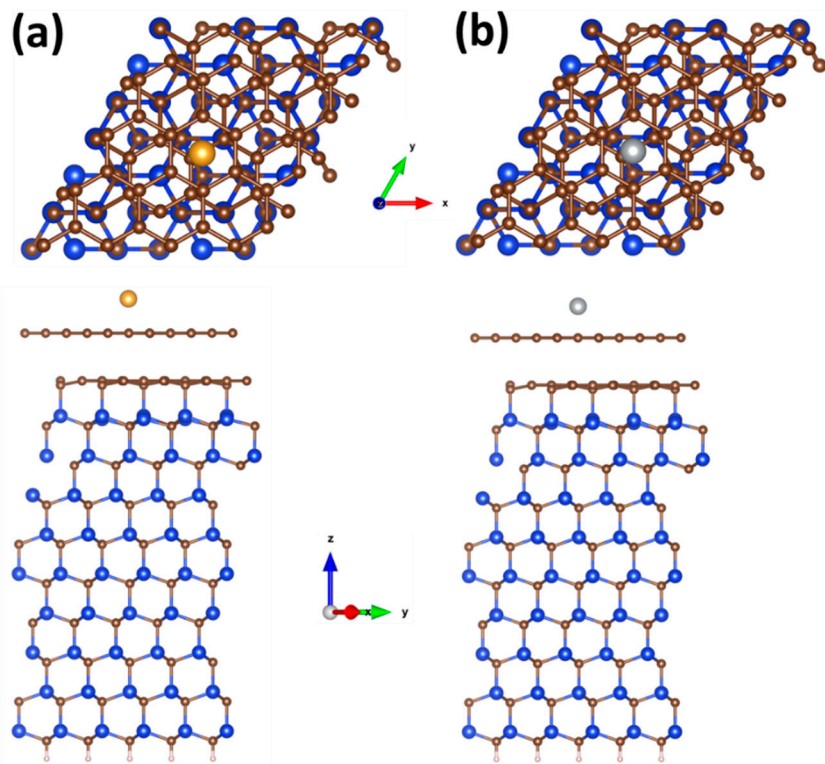

**Figure 7.** (Top view and side view) The optimized geometrical structures of gold (**a**) and silver (**b**) adatoms adsorbed onto monolayer epitaxial graphene. Blue, brown, and whitish balls designate Si, C, and H atoms, respectively. Yellow and gray colors correspond to gold and silver, respectively. This figure has been adapted with permission from [49]. Copyright 2021 WILEY-VCH Verlag GmbH & Co. KGaA, Weinheim, Germany.

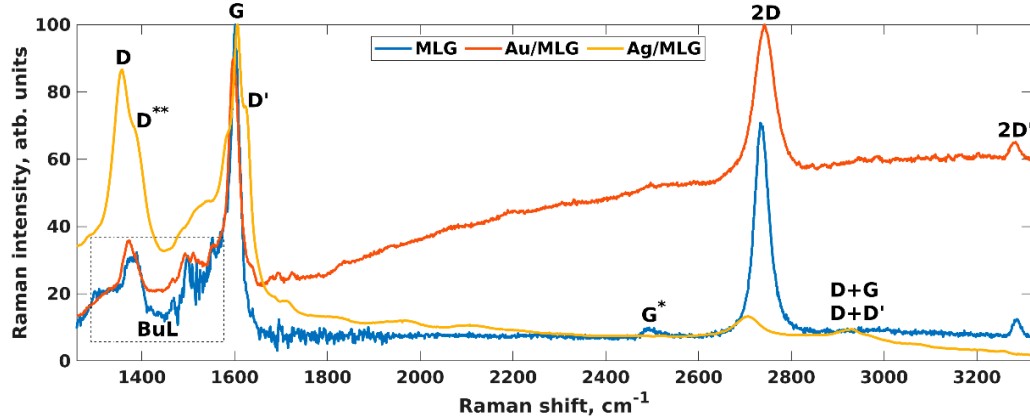

**Figure 8.** Typical Raman spectra of pristine epitaxial graphene, 2 nm-Au-covered epitaxial graphene, and 2 nm-Ag-covered epitaxial graphene, respectively. This figure has been adapted with permission from [49]. Copyright 2021 WILEY-VCH Verlag GmbH & Co. KGaA, Weinheim, Germany.

The effect of magnetron-sputtered Ag film thickness on the morphological evolution during film growth and phonon properties of epitaxial graphene has been thoroughly investigated in [47,48]. As can be seen from Figure 9, only 2 nm-Ag film exhibits island-like morphology with isolated circular islands.

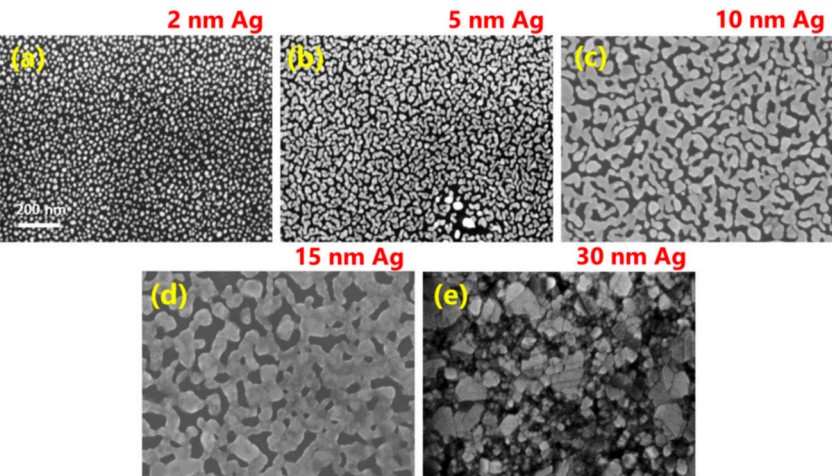

**Figure 9.** SEM images of MLG-supported silver films of different thickness: (**a**) 2 nm, (**b**) 5 nm, (**c**) 10 nm, (**d**) 15 nm, and (**e**) 30 nm, respectively [47]. Copyright 2019 Elsevier B.V., Amsterdam, Netherlands.

Further increase in the nominal thickness of Ag initially led to the appearance of elongated islands (Figure 9b), followed by the formation of an interconnected network (Figure 9c,d) and a continuous silver layer at a thickness of 30 nm (Figure 9e).

Careful analysis of defect-related Raman modes of epitaxial graphene after silver deposition enabled the estimation of the density of generated defects and elucidate their nature [47]. It was revealed that the $D/G$ intensity ratio is very sensitive to alternation of the silver thickness (Figure 10a–d), demonstrating obvious enhancement with increasing of Ag-layer thickness.

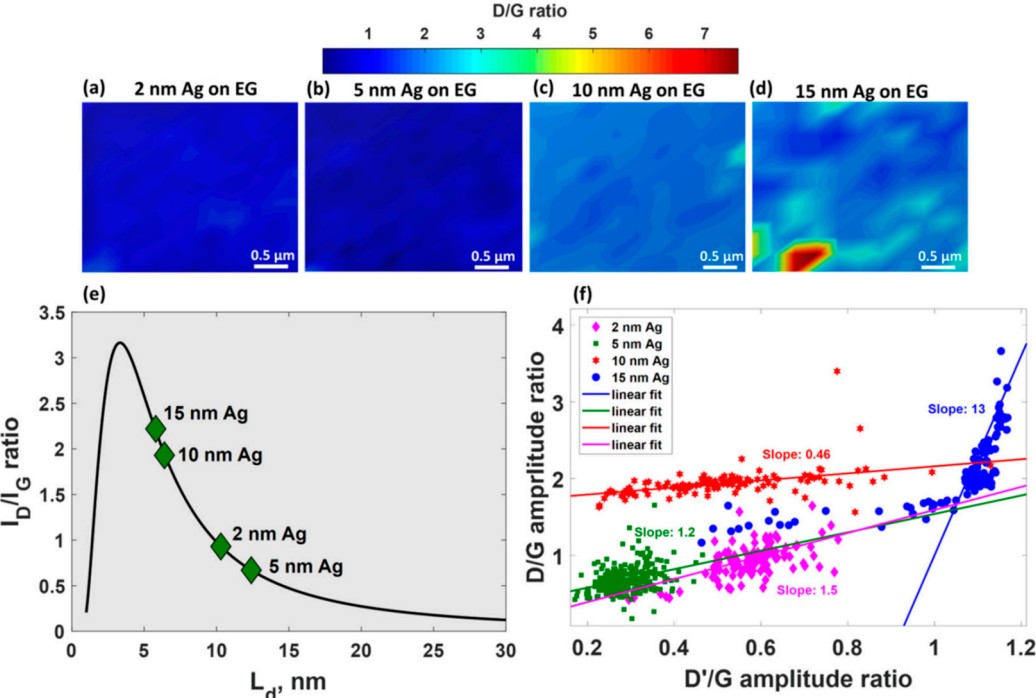

**Figure 10.** SEM images of MLG-supported silver films of different thickness: (**a**) 2 nm, (**b**) 5 nm, (**c**) 10 nm, (**d**) 15 nm, and (**e**) 30 nm, (**f**) Mutual correlation between D/G amplitude ratio on D'/G amplitude ratio for epitaxial graphene after deposition of Ag films with different thicknesses, respectively [47]. Copyright 2019 Elsevier B.V., Amsterdam, Netherlands.

Such behaviour implies two important consequences: (i) reduction in domain size in graphene (Figure 10e) and (ii) increase in density of defects up to $9.5 \times 10^{11}$ cm$^{-2}$ for 15 nm-Ag-decorated epitaxial graphene. The peculiar dependences of $D/G$ amplitude ratio on $D'/G$ amplitude ratio (Figure 10f) for all considered films suggest that the boundary-related defects are preferential defects in the MLG decorated with 2–10 nm Ag films while $sp^3$-type defects dominate in the case of 15 nm-Ag-covered MLG.

It was also shown that the magnetron-sputtered 2 nm- and 5 nm-Ag layers donate electrons to epitaxial graphene, giving rise to a large area uniform *n*-doping of epitaxial graphene ($0.95$–$1.36 \times 10^{13}$ cm$^{-2}$) [48], which is manifested by huge red-shift of $2D$ peak and a very narrow spread of experimental data points in $2D$–$G$ space. Concomitantly, the deposition of thicker Ag films (10 and 15 nm) caused large strain and doping fluctuations.

## 4. Deposition of Metals (Copper, Lead, Mercury, Lithium) on Epitaxial Graphene/4H-SiC by Electroplating

In addition to the dc magnetron sputtering technique, we also employed electrochemical deposition to investigate the early nucleation stages of several selected metals (lead, copper, and mercury) on epitaxial graphene [50–52]. Pb and Hg have been of interest related to toxic metal detection, while Cu attracts a lot of attention as a promising catalyst. A Three-electrode electrochemical cell made of epitaxial graphene as a working electrode, platinum wire as a counter electrode, and an underlying base for the silver-silver chloride reference electrode was designed to perform electrochemical tests. The fundamental mechanism behind this method is a reduction in metal ions at specific potential from different electrolyte solutions on the surface of the epitaxial graphene. The technological maturity of this technique and its simplicity makes it possible to grow ultrathin metal layers under room-temperature conditions quickly, the latter being a big advantage from a practical point of view. Recently, the important role of graphene in minimizing lattice strain and providing conditions for reversible electrodeposition of high-quality metal plates parallel to the electrode has been reported [53]. This is another example suggesting the usability of this method.

Figure 11 demonstrates a summary of cyclic voltammetry measurements performed to study the reduction–oxidation behavior of mercury, copper, and lead.

The knowledge of redox reactions on pristine epitaxial graphene enables the identification of the deposition potential window and can thus be considered as a starting point for further investigation of metal deposition kinetics. The redox behaviour of mercury is quite complicated and slow, since both mercury reduction ($Hg^{2+} + 1e^- = Hg^{1+} + 1e^- = Hg^0$) and oxidation ($Hg^0 - 1e^- = Hg^{1+} - 1e^- = Hg^{2+}$) reactions occur in two steps, which is evidenced by the presence of two reduction and two oxidation peaks (Figure 11a–c) [51]. It was found that the copper electrodeposition on pristine monolayer epitaxial graphene on SiC proceeds via one-step bi-electronic reduction avoiding $Cu^{1+}$ intermediate involvement during the first cycle (Figure 11d) [52]. Concomitantly, during the second cycle, we observed additional oxidation and reduction peaks related to $Cu^{1+}$, which can be attributed to electrochemical processes on $Cu^0$-modified epitaxial graphene. In contrast to mercury and copper electrodeposition, Pb-related redox reactions are one-step and involve the transfer of two electrons [50]. Metallic lead deposition starts at $-0.35$ V (Figure 11e).

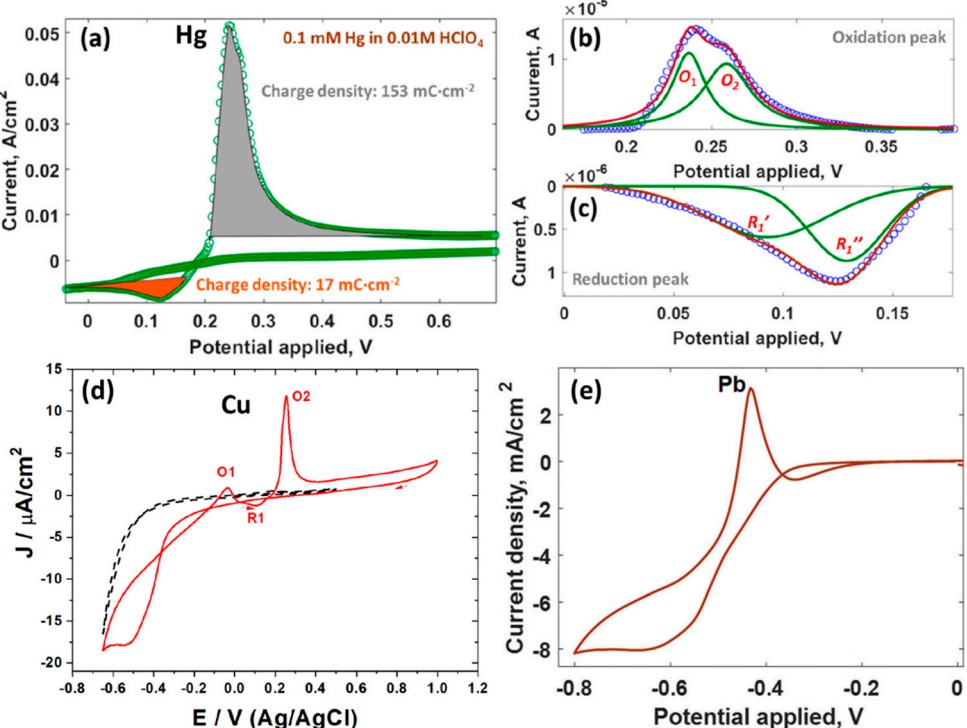

**Figure 11.** Cyclic voltammograms registered at the epitaxial graphene electrode in the presence of mercury [51] (**a**–**c**) (Copyright 2019 by the authors), copper [52] (**d**) (Copyright 2020 by the authors), and lead [50] (**e**) (Copyright the Owner Societies 2018).

To understand the nature of early-stage kinetics of metals on epitaxial graphene in more depth, we then performed chronoamperometry measurements and recorded current transients during metal electrodeposition [50–52]. There are two mechanisms of early-stage nucleation: instantaneous and progressive. The former envisages the fast nucleation on a small number of reactive sites, followed by the subsequent blocking of the nucleation process as soon as the available nucleation sites are depleted. The other mechanism is associated with slow nucleation on a large number of reactive sites since new nuclei keep popping up during the growth process.

The comparison of the experimental current-time curves with theoretical ones predicted by the Hills–Scharifker theory (Figure 12. Note this figure demonstrates only current-time curves collected during Cu and Hg electrodeposition. See the [50], to know more details on Pb case) suggests that the initial kinetics of all considered metals is regulated by the three-dimensional diffusion-controlled instantaneous nucleation mechanism. During the early stages of the electrodeposition process, the epitaxial graphene surface is covered with separated nuclei with discrete diffusion zones, which tend to merge up to the formation continuous metal layer.

Once deposited onto the electrode surface, metal atoms interact with graphene, thereby affecting its physical properties. For example, as was demonstrated in our recent work, electrodeposited lithium species can penetrate beneath both the topmost graphene layer and buffer layer, thereby leading to the formation of bilayer epitaxial graphene with different doping levels per layer [54]. This is confirmed by the apparent *G* peak splitting into two components. Both lead and lithium electrodeposition caused the generation of defects in graphene, which is manifested by an activation of the initially forbidden Raman modes (*D*, *D'* and *D* + *G*) [50,54]. In line with this, an attempt was made to link the metal-graphene interaction strength to the degree of influence of each metal over the phonon properties of epitaxial graphene. It was found that the largest adsorption energy of Li among the other metals is well correlated with the largest red-shift of 2D peak and increased *D/G* ratio, which was observed for lithiated epitaxial graphene. Even a somewhat small

amount of lithium deposited during 5 min has been observed to cause dramatic changes in Raman spectra of epitaxial graphene, which is incomparable to, e.g., the Pb effect. Raman spectroscopy measurements will be performed in a future study to better understand the effect of Hg and Cu electrodeposits on phonon dispersion of epitaxial graphene on SiC.

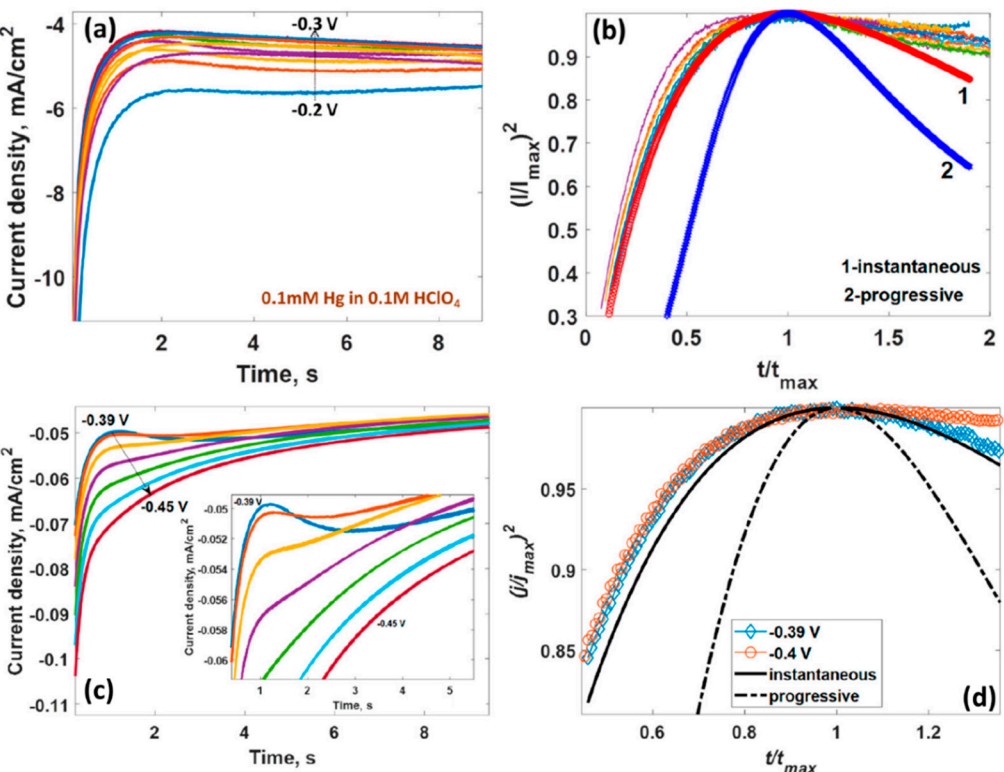

**Figure 12.** Current transients recorded on epitaxial graphene electrodes during mercury [51] (**a**,**b**) (Copyright 2019 by the authors) and copper [52] (**c**,**d**) (Copyright 2020 by the authors) electrodeposition. (**b**,**d**) also encompass the comparison of the normalized current transient curves with theoretical curves determined by the Hills–Scharifker methodology.

## 5. Atomic Layer Deposition of Insulators on Epitaxial Graphene

Apart from the growth of pure metals on epitaxial graphene, the development of controllable and reliable synthesis technology of high-*k* dielectrics (for example, $Al_2O_3$) is of paramount practical importance. It is due to the multifaceted role they play in graphene-based devices as gate insulators in field-effect transistors, tunneling barriers, and protective layers [55–58]. In this context, the integration of epitaxial graphene with insulators can bring this material closer to real electronic applications. In a series of three papers [44,59,60], we presented and critically discussed the atomic layer deposition technique as a promising approach to grow high-quality $Al_2O_3$ layers of different thicknesses. It was revealed that for a deposition temperature of 250 °C, the transition from island-based growth to layer-by-layer growth mode occurs after 40 ALD cycles that correspond to ~2.4 nm (Figure 13).

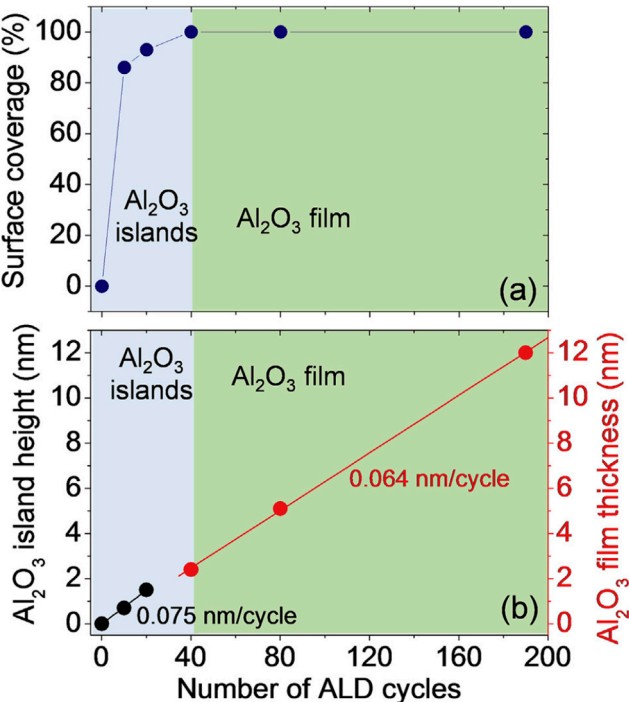

**Figure 13.** (**a**) Surface coverage (in %) of ALD-deposited $Al_2O_3$ on epitaxial graphene/SiC versus a number of ALD cycles. (**b**) Dependences of the height of $Al_2O_3$ islands (left scale) and $Al_2O_3$ film thickness (right scale) on the number of ALD cycles [59]. Copyright 2020 Elsevier Ltd., Amsterdam, Netherlands.

Such a fast transition of the growth mode can be interpreted by the existence of a high density of nucleation centers on the epitaxial graphene surface during early growth stages, which is uncommon for other types of graphene, especially for free-standing exfoliated graphene. This experimental finding is reinforced by the current maps (at a bias $V_{tip}$ = 1 V) measured by conductive-atomic force microscopy (C-AFM) on epitaxial graphene before and after deposition of $Al_2O_3$. The surface of the as-grown (pristine) epitaxial graphene on SiC demonstrates largely uniform conductivity over the scanned area, while the sample subjected to 40 ALD cycles exhibits close-to-zero current, indicating the formation of an insulating film. Obviously, epitaxial graphene on 4H-SiC makes it possible to form highly uniform $Al_2O_3$ thin films without pre-functionalization or seed layers [44]. A cross-sectional TEM image (Figure 14a) of the $Al_2O_3$/epitaxial graphene/SiC structure clearly shows uniform contrast over the whole amorphous $Al_2O_3$ layer, pointing out a uniform density of the material formed during seed-layer-free ALD. AFM measurements (Figure 14b,c) showed that the topography of the formed $Al_2O_3$ layer resembles the terrace-stepped morphology of pristine epitaxial graphene, indicating a small roughness of the ALD-grown film. It is interesting to note that the growth of $Al_2O_3$ on bilayer inclusions was also identified (Figure 14d,e). However, in comparison to the densely packed grains on monolayer epitaxial graphene regions, $Al_2O_3$ on bilayer graphene regions develops into larger grains with obvious separations. Such an effect can be ascribed to higher reactivity of monolayer graphene than that of bilayer or multilayer graphene.

Additional theoretical work has been performed to fully understand the growth mechanism of $Al_2O_3$ on epitaxial graphene [44]. The initial idea was to link the adsorption energies of the precursor molecules (trimethyl aluminium, TMA, and water, $H_2O$) and graphene doping levels. However, no sufficient credible evidence to prove that TMA adsorption depends on the Fermi level position in graphene was provided.

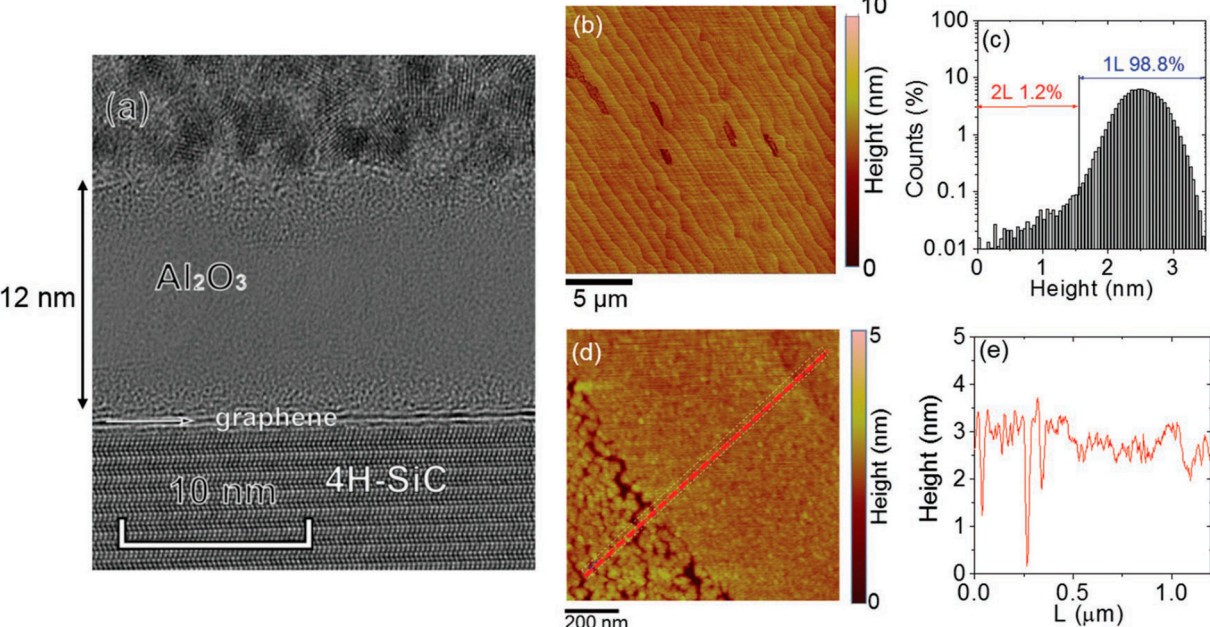

**Figure 14.** (**a**) Cross-sectional TEM image of the 12 nm-thick ALD-grown $Al_2O_3$ film on monolayer epitaxial graphene on SiC. (**b,c**) illustrate the AFM image of the same film corresponding histogram of island height values. (**d,e**) represent a higher resolution AFM image of the same film and height line scan of $Al_2O_3$ at the boundary region between monolayer graphene regions and bilayer graphene inclusion [44]. Copyright 2019 WILEY-VCH Verlag GmbH & Co. KGaA, Weinheim, Germany.

On the other hand, the adsorption energy of water is more sensitive to the doping of graphene and was found to increase with increasing the negative charge on graphene (Figure 5). The stronger the adsorption, the shorter the diffusion path of water molecules. Such locally trapped water molecules may then act as effective reactive sites during subsequent TMA pulses.

Further experiments by using Raman spectroscopy have shown that the ALD process does not cause structural damage of epitaxial graphene, which is evidenced by the absence of defect-related Raman peaks for all investigated samples (Figure 15). The substantial changes related to the blue shift of the 2D peak started to appear only after 80 ALD cycles, and that epitaxial graphene experiences a large compressive strain due to the formation of the continuous film.

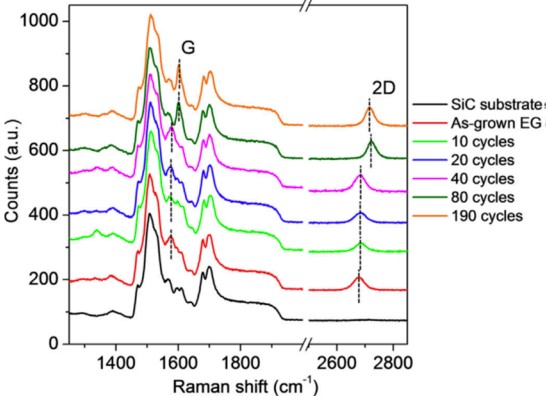

**Figure 15.** Typical Raman spectra of monolayer epitaxial graphene before and after $Al_2O_3$ deposition with 10, 20, 40, 80, and 190 ALD cycles. Raman spectrum of the graphene-free 4H-SiC sample is also shown [59]. Copyright 2020 Elsevier Ltd., Amsterdam, Netherlands.

## 6. Concluding Remarks

We have summarized recent results on the deposition of metals and insulators on epitaxial graphene on 4H-SiC. Three different methods (dc magnetron sputtering, electrodeposition, and atomic layer deposition) were exploited to grow ultra-thin metal and dielectric layers to study early-stage nucleation of selected materials on epitaxial graphene. The comparison of growth morphologies of magnetron-sputtered gold and silver layers supported by DFT calculations and Raman data analysis enabled us to consider a difference in metal-substrate interaction strengths as a candidate explanation of the formation of separated silver islands and fractal-like gold nanostructures at the initial growth stages. We then presented a discussion related to the effect of the grown metal layers on the Raman spectra of epitaxial graphene. Except for gold deposition, the deposition of other metals (Li, Pb, and Ag) caused the generation of defects, which is evidenced by the observation of initially forbidden Raman peaks. The $D/G$ ratio as an important indicator of structural disorder reached the maximal value after Li electrodeposition and hence must be due to the largest number of generated defects. We argued that this finding can be understood in terms of the difference in the adsorption energies of metals and magnitudes of charge transfer from metal to graphene. Three-dimensional diffusion-controlled instantaneous nucleation mechanism was identified as a dominating and probably common mechanism behind initial kinetics of metals during early-stage of the electrodeposition process.

The atomic layer deposition technique recommended itself as a reliable route for seed-layer-free growth of uniform $Al_2O_3$ layers on epitaxial graphene without defects generation. We highlighted the important role which water molecules probably play in providing reactive sites for $Al_2O_3$ formation.

The above-listed observations led to the proposal that the versatility of epitaxial graphene on SiC as a hosting substrate for other materials growth is due to its enhanced reactivity compared to chemically inert free-standing exfoliated graphene. Indeed, numerous experimental results by theoretical predictions indicate that by virtue of its physical and structural properties, epitaxial graphene may contain several different highly reactive regions (step edges, charge-unbalanced electron-hole puddles, strained regions, and edges of bilayer inclusions) that may serve as effective nucleation centres. The cumulative effect of the mentioned regions on the growth kinetics makes the epitaxial graphene on SiC a unique platform for both boosting the development of synthesis technologies of novel hybrid materials and designing high-performance electronic and sensing devices.

**Author Contributions:** Conceptualization, I.S., F.G. and R.Y.; methodology, I.S. and F.G.; software, I.S.; validation, I.S., F.G. and R.Y.; formal analysis, I.S.; investigation, I.S., F.G. and R.Y.; resources, F.G. and R.Y.; data curation, I.S. and F.G.; writing—Original draft preparation, I.S.; writing—Review and editing, F.G. and R.Y.; visualization, I.S.; supervision, R.Y.; project administration, R.Y. and F.G.; funding acquisition, R.Y. and F.G. All authors have read and agreed to the published version of the manuscript.

**Funding:** This research received no external funding.

**Institutional Review Board Statement:** Not applicable.

**Informed Consent Statement:** Not applicable.

**Data Availability Statement:** The data that support the findings of this study are available within the article.

**Acknowledgments:** I.S. acknowledges the support from Ångpanneföreningens Forskningsstiftelse (Grants 16-541 and 21-112). R.Y. acknowledges financial support from VR grant 2018-04962. Financial support by SSF via grant RMA 15-0024 is greatly acknowledged. F.G. acknowledges financial support from MIUR in the framework of the FlagERA project "ETMOS".

**Conflicts of Interest:** The authors declare no conflict of interest. The funders had no role in the design of the study; in the collection, analyses, or interpretation of data; in the writing of the manuscript, or in the decision to publish the results.

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
