# Peer review of "Epitaxial Graphene on 4H-SiC (0001) as a Versatile Platform for Materials Growth: Mini-Review"

_applsci, doi:10.3390/app11135784_

Round 1

Reviewer 1 Report

This is a fine review on an interesting subject, it is well written and easy to follow.

Minor remarks

Ref 51 the name of the journal is missing

The redox behaviour of mercury is quite com- 282
plicated and slow, since both mercury reduction (Hg2+-1e- = Hg1+-1e-= Hg0) and oxidation 283
(Hg0 + 1e- = Hg1+ + 1e- = Hg2+)

+1e- and -1e- should be exchanged

Author Response

Dear Reviewer #1

We use the opportunity to thank the referee for the careful review of our paper and the valuable comments. The changes in the manuscript are marked in yellow, while the answers to the questions appear below in red.

  1. Reviewer’s comment

Ref 51 the name of the journal is missing

  1. Authors’ response

[Ref. 51] is correct.

Shtepliuk, I.; Vagin, M.; Yakimova, R. Insights into the Electrochemical Behavior of Mercury on Graphene/SiC Electrodes. C 2019, 5, 51. https://doi.org/10.3390/c5030051

C is the abbreviation of Journal of Carbon Research published by MDPI.

  1. Reviewer’s comment

The redox behaviour of mercury is quite com- 282

plicated and slow, since both mercury reduction (Hg2+-1e- = Hg1+-1e-= Hg0) and oxidation 283

(Hg0 + 1e- = Hg1+ + 1e- = Hg2+)

+1e- and -1e- should be exchanged

  1. Authors’ response

In the revised version of the manuscript, we made required changes. Indeed, the correct reaction is:

“reduction (Hg2++1e- = Hg1++1e-= Hg0) and oxidation (Hg0 - 1e- = Hg1+ - 1e- = Hg2+)”

Reviewer 2 Report

It is known that at present, including by the method of silicon carbide surface destruction, it has been possible to obtain graphene films of a sufficiently large area and with a sufficiently high structural perfection. However, for the further development of graphene electronics, it is also necessary to work out the processes of deposition of metal and dielectric films on graphene. This mini-review is devoted to these questions. In my opinion, the authors have successfully coped with the task. They analyzed a large number of literature sources and made a detailed analysis of them. The resulting review, in my opinion, will be of interest to both researchers involved in the study of graphene and other 2D materials, and engineers involved in the development of devices based on graphene. I believe that this review can be published in the journal as it is.

Author Response

Dear Reviewer #2

We use the opportunity to thank the referee for the careful review of our paper and the valuable comments.

Reviewer 3 Report

The submitted manuscript entitled "Epitaxial Graphene on 4H-SiC (0001) as a Versatile Platform for Materials Growth" is a timely review on a topic of very strong current interest with graphene epitaxially grown on a semiconductor substrate having a wide bandgap and direct gap, hexagonal silicon carbide 4H-SiC(0001). After describing the key properties of epitaxial graphene as an especially interesting substrate for the deposition of a large variety of materials including noble metals, copper, lead and mercury,  lithium alkali metal, and finally Al2O3 insulator on epitaxial graphene using various appropriate methods of deposition including DC magnetron sputtering, electro-deposition and ALD-atomic layer deposition. Some of the results are supported by DFT (density-fonctionnal theory) calculations and Raman data. The results are very interesting and cover a  broad field of interest including fundamental and applied science.

So this manuscript is especially suitable for publication in Applied Science

Author Response

Dear Reviewer #3,

We use the opportunity to thank the referee for the careful review of our paper and the valuable comments.